# TalkCuts: A Large-Scale Dataset for Multi-Shot Human Speech Video Generation

**Jiaben Chen**[1*]   **Zixin Wang**[1*]   **Ailing Zeng**[2]   **Yang Fu**[1]   **Xueyang Yu**[1]   **Siyuan Cen**[1]
**Julian Tanke**[3]   **Yihang Chen**[4]   **Koichi Saito**[3]   **Yuki Mitsufuji**[3]   **Chuang Gan**[1]

[1]UMass Amherst   [2]Independent Researcher   [3]Sony AI   [4]UC San Diego

## Abstract

In this work, we present *TalkCuts*, a large-scale dataset designed to facilitate the study of multi-shot human speech video generation. Unlike existing datasets that focus on single-shot, static viewpoints, *TalkCuts* offers 164k clips totaling over 500 hours of high-quality human speech videos with diverse camera shots, including close-up, half-body, and full-body views. The dataset includes detailed textual descriptions, 2D keypoints and 3D SMPL-X motion annotations, covering over 10k identities, enabling multimodal learning and evaluation. As a first attempt to showcase the value of the dataset, we present *Orator*, an LLM-guided multimodal generation framework as a simple baseline, where the language model functions as a multi-faceted director, orchestrating detailed specifications for camera transitions, speaker gesticulations, and vocal modulation. This architecture enables the synthesis of coherent long-form videos through our integrated multimodal video generation module. Extensive experiments in both pose-guided and audio-driven settings show that training on *TalkCuts* significantly enhances the cinematographic coherence and visual appeal of generated multi-shot speech videos. We believe *TalkCuts* provides a strong foundation for future work in controllable, multi-shot speech video generation and broader multimodal learning. Project page: https://talkcuts.github.io/.

## 1   Introduction

Human-centric videos permeate modern life across entertainment, communication, and education domains. Although significant advances have been made in synthesizing such videos from multimodal inputs, including speech (Xu et al., 2024a; Cui et al., 2024a,b), 2D keypoints (Hu, 2024; Zhang et al., 2024a; Zhu et al., 2024), and 3D human motion (Zhu et al., 2024), state-of-the-art methods remain limited to single static camera shots, constraining their application to short-form video synthesis.

In contrast, long-form human-centric videos incorporate multiple shots, such as wide angles, closeups, and pans (Lin et al., 2025), to break visual monotony, establish mood, and emphasize key speech elements. This requires orchestrating several interacting systems: the speaker's articulation, gestures, and movement within the scene, alongside dynamic camera work that responds appropriately to spoken content. Previous human video generation models focus on generating short and single-shot videos, instead of long and multi-shot videos with consistent human appearance (Liu et al., 2024; Tian et al., 2024; Xu et al., 2024b; Hu et al., 2023; Corona et al., 2024; Kong et al., 2025). Meanwhile, existing human-centric video datasets are inadequate for this task, as they primarily consist of short-form content such as TikTok dances (Jafarian and Park, 2021) or fashion videos (Zablotskaia et al., 2019), typically offering at most one additional modality (e.g., speech or 2D keypoints) and remaining relatively limited in scale.

To address this gap, we present *TalkCuts*, a large-scale dataset specifically curated for long-form human speech video generation with dynamic camera shots. This comprehensive collection features

39th Conference on Neural Information Processing Systems (NeurIPS 2025) Track on Datasets and Benchmarks.

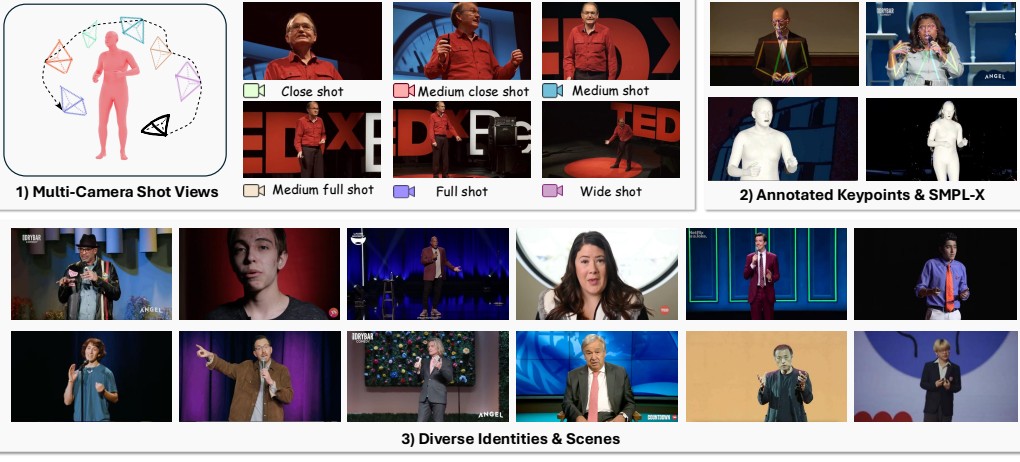

Figure 1: **Overview of the TalkCuts Dataset.** The dataset features (1) diverse camera shot types (e.g., close-up, half-body, full-body), (2) annotations for 2D keypoints and 3D SMPL-X motion, and (3) a wide range of speaker identities spanning various ethnicities, body types, and age groups.

videos from talk shows, TED talks, stand-up comedy, and diverse speech scenarios, encompassing more than 10,000 unique speaker identities. Each video contains multiple camera shots and is annotated with 2D whole-body keypoints, 3D SMPL-X estimations and textual descriptions. With 1080p resolution and over 500 hours of footage, *TalkCuts* represents the largest publicly available dataset of its kind. All content has undergone rigorous filtering and annotation to ensure high quality, providing a robust resource for training and evaluating models that generate realistic, multi-shot speech videos in dynamic settings. We compare *TalkCuts* with recent human video datasets in Table 1 and present a visual overview in Figure 1.

Based on this dataset, we propose a novel task: given a script and reference images, generate coherent multi-shot human speech videos. We evaluate this long-form human-centric video generation in two domains: camera shot transition and audio-driven human video generation. We also provide results under a pose-guided video generation setting to further demonstrate the effectiveness of *TalkCuts*. Our experiments show that models trained on *TalkCuts* consistently outperform existing baselines across multiple metrics, including shot coherence, motion quality, and identity preservation.

Table 1: **Comparison of existing public datasets** for pose-guided video generation (top) and audio-to-gesture generation (bottom), categorized by meta information, modality, and camera details.

| Dataset | Meta Information | | | | | Modality | | | Camera |
|---|---|---|---|---|---|---|---|---|---|
| | Clips | Frames | Resolution | Hours | ID | 2D Annot. | 3D Annot. | Audio | Shots |
| **Pose-guided Generation Datasets** | | | | | | | | | |
| TikTok (Jafarian and Park, 2021) | 340 | 93k | 604x1080 | 1.03 | ≈300 | ✗ | ✗ | ✓ | Single |
| TED Talks (Siarohin et al., 2021) | 1322 | 197k | 384x384 | - | 173 | ✗ | ✗ | ✓ | Single |
| UBC-Fashion (Zablotskaia et al., 2019) | 500 | 192k | 720x964 | 2 | ≈600 | DWPose | ✗ | ✗ | Single |
| **Audio-to-gesture Generation Datasets** | | | | | | | | | |
| Speech2Gesture (Ginosar et al., 2019) | - | - | - | 144 | 10 | OpenPose | ✗ | ✓ | Single |
| UBody (Lin et al., 2023) | - | 1051k | - | 11.7 | - | DWPose | SMPL-X | ✗ | Single |
| TalkSHOW (Yi et al., 2023) | 17k | - | - | 38.6 | 4 | ✓ | SMPL-X | ✓ | Single |
| BEAT2 (Liu et al., 2023) | - | 32M | 1080P | 76 | 30 | ✓ | SMPL-X | ✓ | Single |
| **TalkCuts (Ours)** | 164k | 57M | 1080P | 507 | 11k+ | DWPose | SMPL-X | ✓ | Multi |

As a baseline, we introduce *Orator*, an end-to-end system for automatically synthesizing long-form speech videos with dynamic camera shots, as is illustrated in Figure 2. The system comprises two key components: a DirectorLLM and a multi-modal generation module. The DirectorLLM is an LLM-driven multi-role director that orchestrates the entire generation process through textual guidance. It integrates speech content, camera transitions (*e.g.*, close-up, medium, or wide shots) based on emotional flow, and gesture descriptions aligned with the speaker's actions. Additionally, it provides vocal delivery instructions to modulate tone, emotion, and pacing. The multi-modal generation module translates text into long-form speech videos with natural camera transitions, synchronized

gestures, and dynamic vocal delivery. The module includes SpeechGen, which processes input text along with LLM-generated audio instructions to produce synchronized speech, and VideoGen, which integrates the generated audio, input reference images, and motion instructions from the LLM using a video diffusion model to generate the final video.

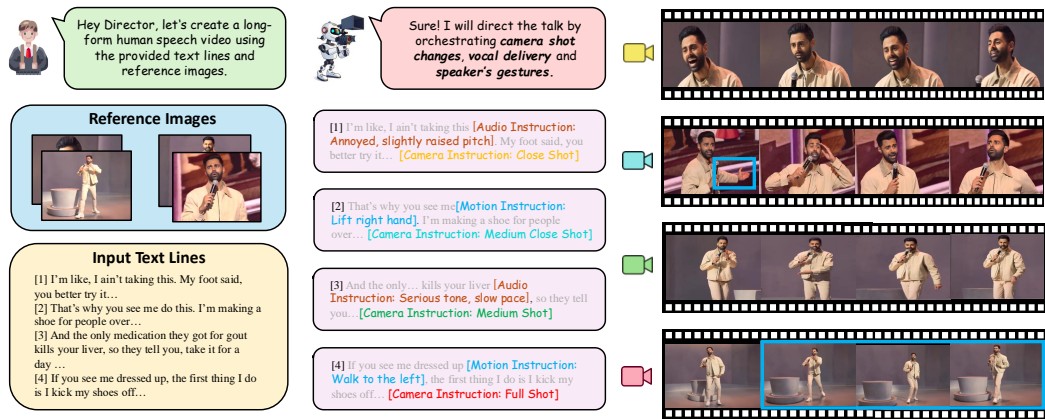

Figure 2: **Multi-shot speech video generation.** We propose *Orator*, a fully automated system that generates human speech videos with dynamic camera shots. By organically integrating multiple modules, a DirectorLLM directs camera transitions, gestures, and audio instructions, delivering coherent and engaging multi-shot speech videos.

In summary, this paper introduces the novel task of speech-driven video generation with dynamic camera shots across multiple scenarios (head, half-body, and full-body views). We present *TalkCuts*, the first large-scale dataset specifically designed for long-form human speech generation, featuring diverse scenarios and comprehensive annotations including multi-shot camera transitions and 3D SMPL-X motion data. Furthermore, we propose *Orator*, an automated pipeline for fine-grained video generation that maintains visual identity consistency across camera transitions through a multimodal generation system guided by DirectorLLM. Extensive experiments validate the effectiveness of *TalkCuts* in enabling realistic and coherent multi-shot speech video generation.

## 2 Related Works

**Human Video Datasets.** Recently, various datasets derived from public platforms such as TikTok and YouTube have been introduced to advance human video generation research. For example, the TikTok dataset (Jafarian and Park, 2021) includes 340 short video clips, each lasting 10-15 seconds, primarily featuring dancing humans. However, these datasets are limited in both scale and quality. To overcome these limitations, several synthetic datasets (Varol et al., 2017; Patel et al., 2021; Cai et al., 2021; Yang et al., 2023a) have been developed, significantly enhancing the diversity of backgrounds and the scale of training data. Besides, the importance of multimodal cues has become increasingly evident. HumanVid (Wang et al., 2024), comprising over 50 million frames, is annotated using mature and widely adopted tools, offering a valuable resource for large-scale learning. However, existing datasets remain limited in several aspects: they are largely constrained by identity-specific annotations, lack comprehensive multi-shot labeling, and do not provide standardized benchmarks for evaluating shot transitions. To address these limitations, we introduce *TalkCuts*, a benchmark specifically designed for multi-shot video generation, featuring over 10,000 unique identities to enable scalable evaluation.

**Audio-driven Human Video Generation.** In audio-driven video generation, prior works (Sun et al., 2023; Zhang et al., 2023) mainly focus on achieving accurate lip synchronization and semantic alignment with speech. To expand the generated region, Vlogger (Corona et al., 2024) synthesizes half-body human videos. EchoMimicV2 (Meng et al., 2024) supports combinations of both audios and selected facial landmarks and in the meantime enhances half-body details. Nevertheless, such models are constrained to static background settings and are not equipped to effectively model or generate dynamic backgrounds. Upon these models, Hallo3 (Cui et al., 2024b) adopts a two-stage training framework that leverages LLMs to enrich textual descriptions, thereby enhancing the model's ability to comprehend scene context and generate coherent videos with dynamic backgrounds. Despite

these advances, existing models remain incapable of generating coherent cross-shot and multi-shot video sequences. To the best of our knowledge, our proposed baseline is the first to achieve structured and coherent multi-shot speech video generation.

# 3   TalkCuts Dataset

We introduce *TalkCuts*, a large-scale human video dataset specifically designed for speech scenarios such as TED talks and talkshows. *TalkCuts* provides high-resolution speech videos with varying camera shots, and includes diverse modalities such as synchronized texts, audio, 2D keypoints, 3D SMPL-X parameters, and video descriptions, enabling comprehensive multimodal training and evaluation for multi-shot speech video generation. This dataset provides a comprehensive benchmark for future research, facilitating further improvements in human video generation.

## 3.1   Data Curation

**Data Collection.** We performed keyword searches targeting different speech scenarios on YouTube to crawl copyright-free, high-resolution real-world videos. Manual filtering was applied to remove low-quality or irrelevant content and only videos featuring a clearly visible human speaker with corresponding speech audio were retained.

**Data Filtering and 2D Keypoints Detection.** We use PySceneDetect (Castellano) to segment each video into multiple clips based on scene transitions. To ensure high-quality clips, we apply RTMDet (Lyu et al., 2022) from MMDetection (Chen et al., 2019) for human detection. Clips are filtered out if no human or multiple humans are detected, or if the bounding box is too small. For the remaining clips, we apply DWPose (Yang et al., 2023b) for human pose estimation to obtain the COCO whole body pose with 133 keypoints. Final filtering is based on the head keypoints confidence scores, discarding clips with low scores for key facial points.

**Data Statistics.** Our dataset contains over 500 hours of video, with 164K clips and 57M frames, featuring more than 10K unique speaker identities, all in 1080p resolution. Table 1 provides a comprehensive comparison of our dataset with existing speech video datasets, highlighting its scale, diversity, and rich annotations, including multi-camera-shots and 3D SMPL-X motion data. Additionally, as shown in Fig. 1, our dataset captures a wide range of speech scenarios (e.g., TED talk, stand-up comedy, presentation, lecture, interview, talkshow and so on), featuring diverse speaker demographics (in terms of race, body type, and age) and various camera shots for each identity, making it suitable for training and evaluating multi-shot speech video generation models.

## 3.2   Data Annotation

**Camera Shots Definition and Annotation.** In our paper, we classify camera shots into six types: Close-Up (CU), Medium Close-Up (MCU), Medium Shot (MS), Medium Full Shot (MFS), Full Shot (FS), and Wide Shot (WS) based on established cinematographic principles (shown in Figure 1), as outlined by (Brown, 2016). This classification allows for capturing various visual details, from intimate facial expressions to contextualizing the subject within their environment. To annotate each clip with a corresponding shot type, we analyze the 2D keypoints detected for each segment and determine the visible body parts, such as head, torso, or full body, and then map them to the appropriate shot category.

**3D SMPL-X Annotation.** We adopt the SMPL-X (Pavlakos et al., 2019) model to represent 3D human motion. For a given T-frame video clip, the corresponding pose states $\mathcal{P}$ are represented as: $\mathcal{P} = \{\mathcal{P}_f, \mathcal{P}_b, \mathcal{P}_h, \zeta, \epsilon\}$, where $\mathcal{P}_f \in \mathbb{R}^{T \times 3}$, $\mathcal{P}_b \in \mathbb{R}^{T \times 63}$, and $\mathcal{P}_h \in \mathbb{R}^{T \times 90}$ represent the jaw poses, body poses, and hand poses, respectively. $\zeta \in \mathbb{R}^{T \times 10}$ and $\epsilon \in \mathbb{R}^{T \times 3}$ denote the facial expressions and global translation. We initially use the state-of-the-art method SMPLer-X (Cai et al., 2024) to estimate the whole-body motion sequence $\mathcal{P}$, but observed limitations in the accuracy of face and hand parameters, specifically $\mathcal{P}_f$, $\zeta$, and $\mathcal{P}_h$. To address this, we refine the hand poses $\mathcal{P}'_h$ using HaMeR (Pavlakos et al., 2024), and improve the jaw poses $\mathcal{P}'_f$ and facial expressions $\zeta'$ using EMOCA (Danecek et al., 2022; Feng et al., 2021). We then combine the refined $\mathcal{P}'_f$, $\zeta'$, and $\mathcal{P}'_h$ into the original pose prediction $\mathcal{P}$ to obtain the final high-quality motion estimation $\mathcal{P}'$.

**Video Textual Description Annotation.** We use the latest Qwen2.5-VL (Bai et al., 2025) to generate textual descriptions for each video. Each clip is uniformly sampled into 16 frames. For each clip, the

VLM is prompted to summarize the content with a focus on: 1) Detailed descriptions of the speaker's head movements, hand gestures, and body posture changes; 2) The speaker's facial expressions and their variations (e.g., smiles, frowns, raised eyebrows), especially those conveying emotions and delivery style; 3) Camera shot types (e.g., close-up, medium shot, full shot) and any visible camera movements (e.g., zoom-in, pan left) and 4) Detailed descriptions of the environmental background.

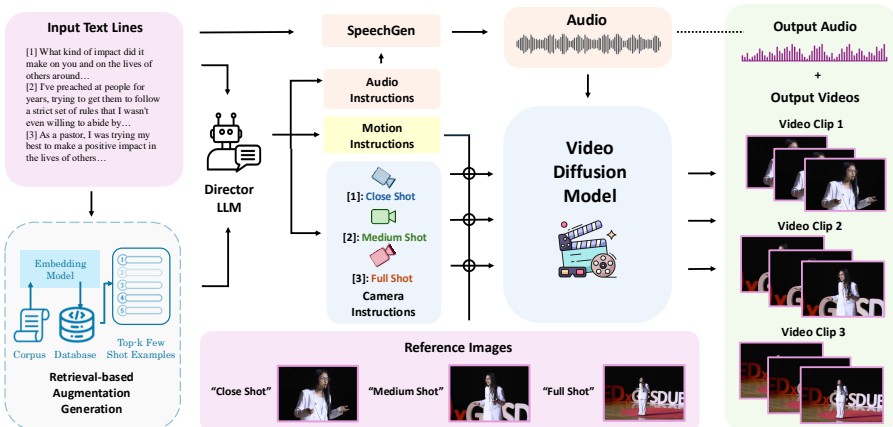

Figure 3: **Pipeline of Orator.** The DirectorLLM processes the input script to generate instructions for camera shots, motion, and audio. These guide the multi-modal video generation model to produce the final long-form speech video with natural transitions and gestures.

## 3.3 Orator: A Simple Baseline for Multi-Shot Speech Video Generation

**Overall Framework.** To address the task of multi-shot human speech video generation, we introduce a simple baseline, intended to serve as a reference point for subsequent research. The overall framework of *Orator* is shown in Figure 3, which consists of a DirectorLLM and a Multimodal Video Generation Module. Given an input speech script $S$ and a set of reference images $\{I_k\}_{k=1}^K$ from different camera angles, our framework aims to automatically generate a long-form speech video $V$ with natural camera shot transitions. First, the DirectorLLM takes the speech script $S$ as input and generates camera shot instructions $\{T_i^c\}_{i=1}^N$, specifying when and how to transition between shots. These instructions segment the script into $N$ segments $\{S_i\}_{i=1}^N$, each corresponding to a distinct shot. For each segment, the DirectorLLM additionally produces motion instructions $\{T_i^m\}_{i=1}^N$ for the speaker's expression, gestures and body movements, along with audio instructions $\{T_i^a\}_{i=1}^N$ for vocal delivery, such as tone, pace and emphasis.

These multi-modal instructions $\{T_i^c\}_{i=1}^N$, $\{T_i^m\}_{i=1}^N$, and $\{T_i^a\}_{i=1}^N$ are then passed to the generation module. The SpeechGen module processes each text segment $S_i$ with the audio instructions $T_i^a$ to generate the vocal output $A_i$. Then, following the camera shot plan $\{T_i^c\}_{i=1}^N$, the VideoGen module takes the generated speech audio $\{A_i\}_{i=1}^N$ and the reference images $\{I_k\}_{k=1}^K$, and incorporates the motion instructions $\{T_i^m\}_{i=1}^N$ to synthesize the final video $V$ via a video diffusion model.

### 3.3.1 DirectorLLM: A Multi-Role Video Director

In this section, we describe the DirectorLLM's role in orchestrating the key elements of video generation: camera shot planning, speaker gesture control, and vocal delivery guidance.

**LLM as Camera Shots Planner.** The DirectorLLM analyzes the speech script $S$ and generates camera shot instructions $\{T_i^c\}_{i=1}^N$, which are then utilized to segment the script into $N$ segments $\{S_i\}_{i=1}^N$ corresponding to different camera shots. These shot instructions determine the optimal camera angle transitions based on the narrative structure, emotional flow, and key emphasis points within the speech. The LLM selects camera shots based on narrative structure, emotional intensity, and key moments in the script, recommending shot transitions like *"close-up"* (close_up_shot) during the emotional highlights and *"wide shots"* (wide_shot) for contextual emphasis. In our approach to automatic shot division, we employ a Retrieval-Augmented Generation (RAG)-based method (Guu et al., 2020; Lewis et al., 2020), leveraging GPT-4o (Achiam et al., 2023) to produce

shot transitions $\{T_i^c\}_{i=1}^N$ for video content based on speech. The process begins by extracting text embeddings $E(S)$ from the input speech $S$ using a text-embedding model. We then compute the cosine similarity between the input embeddings $E(S)$ and a pre-computed set of embeddings $\{E(S_j)\}_{j=1}^M$ from our training dataset, the Shot Division Corpus (SDC), which contains speech segments paired with ground-truth shot transitions $\{T_j^c\}_{j=1}^M$. Using this, we retrieve the top-5 most similar speech segments based on the cosine similarity. These retrieved examples $\{S_j\}_{j=1}^5$ and their corresponding shot transitions $\{T_j^c\}_{j=1}^5$, are used as few-shot prompts for GPT-4o (Achiam et al., 2023). Given these contextually relevant examples, GPT-4o generates a shot transition plan $\{T_i^c\}_{i=1}^N$ for the input speech $S$. This approach enables the model to adapt its predictions by learning from past similar examples, capturing the nuanced relationship between speech content and shot division.

**LLM as Motion Instructor.** The DirectorLLM also acts as a motion planner, guiding the speaker's body language, gestures, and movement on stage to enhance the delivery of the speech. For each speech segment $S_i$, the LLM motion instructions $\{T_i^m\}_{i=1}^N$, tailored to the content and emotional tone of the speech. For gestures, the LLM analyzes key points of emphasis and emotion to suggest actions like *"raise right hand"* (gesture_raise_right_hand) or *"open arms"* (gesture_open_arms) during moments of intensity. For more reflective segments, it might recommend subtler movements like *"fold hands"* (gesture_fold_hands). In addition to gestures, the LLM provides instructions for stage movement. Based on the flow of the speech, the LLM suggests where and when the speaker should move on stage, suggesting instructions such as *"move left"* (move_left) or *"step forward"* (step_forward) to maintain a dynamic presence.

**LLM as Voice Delivery Instructor.** *DirectorLLM* generates fine-grained vocal instructions, including intonation, pitch, pace, and emotion, to guide speaker delivery and enhance engagement. For each speech segment $S_i$, it outputs vocal instructions $T_i^{a N}_{i=1}$ aligned with emotional tone and context. Sentence-level control is achieved via prompt-based cues (e.g., tone_calm, pitch_low, slow_pace), such as *"calm tone and lower pitch"* for introductions or *"slow down for emphasis"* in critical moments. The LLM can also leverage token-based control for fine-grained adjustments by inserting word-level emphasis, breathing, or laughter tokens. For instance, it can emphasize key terms: *"The only medication they have for gout kills your liver"* or add realism with [breath] or [laughter] tokens: *"I'm like, I ain't taking this... [breath] My foot said, you better try it."*. These multi-level controls allow the LLM to dynamically adapt vocal delivery to the speech's emotional rhythm, resulting in more expressive and natural speech-driven videos.

### 3.3.2 Multimodal Video Generation

To enable the automatic generation of long-form speech videos with natural camera transitions, we propose a multimodal video generation pipeline composed of two main modules: SpeechGen, responsible for generating synchronized speech audio, and VideoGen, responsible for synthesizing the final video. Both modules operate under the guidance of instructions provided by DirectorLLM.

**SpeechGen.** The SpeechGen module is responsible for generating expressive speech audio based on the vocal instructions provided by the DirectorLLM. After receiving the vocal instructions $\{T_i^a\}_{i=1}^N$, which specify the tone, pitch, pace, and pauses for each speech segment $S_i$, the SpeechGen module processes the input text lines $S_i$ and generates corresponding audio output $A_i$. We utilize the text-to-speech model CosyVoice (Du et al., 2024), which is instruction fine-tuned (Ji et al., 2024) for enhanced controllability. The model allows for sentence-level adjustments such as emotion, speaking rate, and pitch, as well as token-level controls to insert elements like laughter, breaths, and word emphasis. The SpeechGen module seamlessly integrates these controls from the DirectorLLM, with sentence-level prompts guiding the overall tone and pacing, and tokens like  for emphasis and [breath] for natural pauses. This combined approach ensures that the generated audio synchronizes with the speech content and emotional flow.

**VideoGen.** The VideoGen module synthesizes human speech videos based on reference images $\{I_k\}_{k=1}^K$, speech audio $\{A_i\}_{i=1}^N$ from the SpeechGen module, and motion instructions $\{T_i^m\}_{i=1}^N$. The objective is to generate visually coherent videos that accurately align with the speech audio while maintaining the identity and visual consistency of the speaker across different camera shots.

To achieve this, we build upon CogVideoX Yang et al. (2024b), a powerful transformer-based diffusion model that utilizes a 3D VAE to enhance video quality and ensure narrative coherence. By leveraging this pretrained model, we significantly reduce both data requirements and computational

costs. To enable audio-driven human video generation, we integrate the latest Hallo3 (Cui et al., 2024b) architecture, a two-stage DiT-based framework. In the first stage, we enhance identity preservation by injecting reference features into the video generation process. A T5-based text encoder (Raffel et al., 2020) encodes the motion instructions $T_i^m$ into text embeddings, which, along with identity features extracted from the reference image using a 3D causal VAE, are processed by an identity reference network. This network generates reference features, which are then injected into the 3D attention blocks of the denoising network. Additionally, cross-attention layers take facical embedding extracted from the reference image $I_{k_i}$ via InsightFace (DeepInsight, 2024) to further refine identity consistency across frames. In the second stage, the model is fine-tuned for audio-driven generation by conditioning the denoising network on Wav2Vec extracted (Schneider et al., 2019) speech embeddings, which are fed into audio attention modules via cross attention. Finally, for each video segment, we generate individual video clips $V_i$ by combining the corresponding speech audio $A_i$ and the reference image $I_{k_i}$. The final long-form speech video $V$ with different camera shots is obtained by concatenating all the generated video clips $\{V_i\}_{i=1}^N$.

While the pretrained models offer strong identity preservation and synchronization, they exhibit limitations when applied to multi-shot settings, particularly in maintaining fine details in hand regions and object interactions. Additionally, since the original model was primarily trained on portrait scenarios and upper-body shots, their performance degrades significantly when handling half-body, full-body, and side-angle views. To address these issues, we fine-tune the model with two stages on our dataset, specifically adapting the model to speech videos with various camera views.

## 4    Experiments

To evaluate the effectiveness of the *TalkCuts* dataset, we apply the previously introduced *Orator* baseline across two tasks: LLM-guided camera shot transitions (Sec. 4.1) and audio-driven human video generation (Sec. 4.2). In addition, we provide results under a pose-guided video generation setting (Sec. 4.3) to further illustrate the dataset's utility. These experiments demonstrate that *TalkCuts* supports both audio- and pose-driven speech video generation, highlighting its value for advancing multi-shot human video synthesis.

### 4.1    LLM-guided Camera Shot Transitions

**Metrics.** We assess shot planning accuracy using three key metrics: IoU (Intersection over Union), measuring the overlap between predicted and ground truth shot boundaries (higher IoU indicates better alignment); Accuracy, reflecting the percentage of correctly predicted shot types; and Shot Matching Accuracy (SMA), which evaluates how consistently the predicted shot types match the ground truth at specific time intervals.

**Baselines.** We compare several models: GPT-4o (Achiam et al., 2023), LLaMA 3.1-8B-Instruct (Dubey et al., 2024), Qwen 2.5 (Yang et al., 2024a) and Snowflake-Embed (Merrick et al., 2024). For GPT-4o, we evaluate three setups: RAG-fewshot, random-fewshot, and zeroshot. For the RAG-fewshot setup, we utilized text-embedding-3-small and FAISS (Douze et al., 2024) to retrieve the five most similar examples from the training set to serve as few-shot samples. In contrast, for the random-fewshot setup, we randomly selected five examples from the training set. LLaMA 3.1 (Dubey et al., 2024) and Qwen 2.5 (Yang et al., 2024a) are evaluated using similar setups, with fine-tuning performed using LoRA (Hu et al., 2021). Snowflake-Embed, being an embedding model, required the addition of a linear classification head to function as a classifier.

Table 2: **Quantitative results of LLM-guided camera shot transitions.** Z.S. refers to zershot , R.F. refers to random fewshot and Tune refers to fine tune. Best result is shown in **bold** and second best in underline.

| Method | Accuracy↑ | SMA↑ | IOU↑ |
|---|---|---|---|
| Embedding Model | 35.60% | 30.42% | 35.60% |
| Llama 3.1 8B Z.S. | 20.41% | 23.72% | 10.50% |
| Llama 3.1 8B R.F. | 24.63% | 44.01% | 13.28% |
| Llama 3.1 8B RAG | 21.65% | 47.15% | 15.33 % |
| Llama 3.1 8B Tune | **79.09%** | 49.40% | 30.06% |
| Qwen 2.5 7B Z.S | 26.59% | 11.81 % | 19.79% |
| Qwen 2.5 7B RAG | 46.16% | 13.93% | 26.39% |
| GPT-4o Z.S. | 64.34% | 48.34% | 40.59% |
| GPT-4o R.F. | 67.50% | 58.12% | 42.46% |
| GPT-4o RAG (Ours) | 70.66% | **64.06%** | **48.10%** |

**Result Analysis.** We present the comparison between different baselines in Table 2. The embedding model serves as a baseline and shows limited performance without contextual understanding. IoU and SMA values are observed to be better indicators of alignment between the predicted and ground truth shot boundaries compared to accuracy, as high accuracy may be due to overfitting, making Qwen a

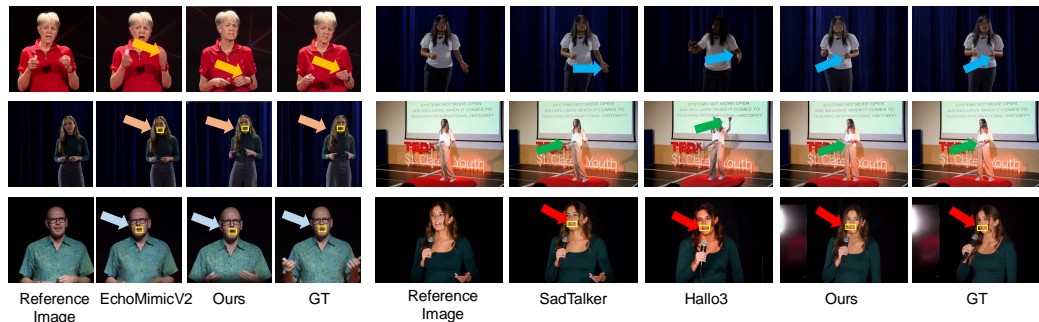

| Reference Image | EchoMimicV2 | Ours | GT | Reference Image | SadTalker | Hallo3 | Ours | GT |

Figure 4: **Qualitative Comparison of Human Video Generation Results.** We compare our results with baseline models across close-up, medium, and full-body shots. Artifacts in baseline outputs, such as facial distortions, motion blur, mismatched hand gestures, and lip-sync inconsistencies, are highlighted with arrows and bounding boxes. Our model produces more realistic results across all shot types, maintaining visual fidelity and smoother transitions compared to the baselines.

suboptimal option. For SMA and IoU, both the Llama (Dubey et al., 2024) and GPT-4o (Achiam et al., 2023) RAG models outperform random-fewshot, indicating that selecting relevant examples in our data corpus improves shot planning performance. It is worth noting that the fine-tuned LLaMA model does not achieve a higher IoU than the Embedding Model, but its SMA is significantly better. This suggests that the fine-tuned Llama model has learned some contextual information. On the other hand, GPT-4o (Achiam et al., 2023), although slightly inferior to the fine-tuned Llama (Dubey et al., 2024) in terms of accuracy, shows much higher SMA and IoU, making it the final chosen model.

Table 3: **Quantitative Comparison for audio-driven human speech video generation.** Best result is shown in **bold** and the second-best result is shown in underline.

| Method | Video Generation Quality | | | | | Lip Sync. | | Consistency & Dynamic Degree | | | |
|---|---|---|---|---|---|---|---|---|---|---|---|
| | FID↓ | FVD↓ | PSNR ↑ | SSIM ↑ | LPIPS ↓ | Sync-C ↑ | Sync-D ↓ | Sub. Cons. ↑ | Back. Cons. ↑ | Dynamic Degree ↑ | Motion Smooth. ↑ |
| SadTalker | 159.78 | 926.01 | 12.46 | 0.356 | 0.572 | **4.57** | 8.89 | 91.72% | **98.99%** | 19.42% | **99.81%** |
| EchoMimicV2 | 177.22 | 1770.22 | 14.88 | 0.582 | 0.511 | 1.94 | 9.65 | 88.61% | 93.25% | 87.10% | 99.30% |
| Hallo3 | 57.28 | 855.84 | 19.22 | 0.644 | 0.215 | 3.84 | 9.76 | 95.81% | 94.33% | 54.84% | 99.45% |
| Ours | **45.86** | **622.76** | **20.61** | **0.708** | **0.198** | 4.35 | **8.32** | **96.24%** | 95.42% | 68.48% | 99.63% |

## 4.2 Audio-driven Human Video Generation

**Metrics.** We evaluate the generation quality across multiple dimensions. To assess overall video quality, we compute FID (Heusel et al., 2017), FVD (Unterthiner et al., 2019b), PSNR, SSIM (Wang et al., 2004), and LPIPS (Zhang et al., 2018). For audio-lip synchronization, we employ SyncNet (Prajwal et al., 2020) to measure Sync-C and Sync-D scores. Additionally, we incorporate VBench (Huang et al., 2024a,b) metrics for a more comprehensive video quality assessment. Subject consistency is evaluated using DINO feature similarity (Caron et al., 2021), while background consistency is measured via CLIP feature similarity (Radford et al., 2021). We further assess motion smoothness by leveraging motion priors from a video frame interpolation model (Li et al., 2023) and quantify the degree of motion dynamics using RAFT optical flow (Teed and Deng, 2020).

**Baselines.** We compare our trained model against recent state-of-the-art publicly available methods for speech-driven human video generation, including SadTalker (Zhang et al., 2023), EchoMimicV2 (Meng et al., 2024), and Hallo3 (Cui et al., 2024b).

**Evaluation Benchmark.** We provide a test set of 50 video clips from our proposed *TalkCuts* dataset, featuring diverse identities and varying camera shot angles for comprehensive evaluation.

**Result Analysis.** As shown in Table 3, our model, trained on the *TalkCuts* dataset with its diverse range of identities and dynamic camera shots, achieves the highest scores in overall video generation quality. For audio-lip synchronization, our model consistently ranks among the top, demonstrating strong alignment between speech and visual articulation. Additionally, across VBench metrics, including subject consistency, background stability, motion smoothness, and dynamic expressiveness, our approach outperforms existing baselines in several key aspects, securing either the highest or second-highest scores. These results highlight the effectiveness of our framework in generating high-fidelity, temporally coherent, and expressive speech videos. Figure 4 presents a qualitative

comparison with previous SOTA methods. We observe that existing models exhibit significant limitations: EchoMimicV2 struggles with accurate lip synchronization, often producing misaligned mouth movements. Hallo3 suffers from motion blur and poor hand region details. In contrast, our method produces natural gestures, detailed hand renderings, and strong identity preservation.

### 4.3 Pose-guided Human Video Generation

**Metrics.** We assess the generation quality across three dimensions: 1) Single-frame image quality using SSIM, PSNR, LPIPS, and FID; 2) Video quality measured by FVD (Unterthiner et al., 2019a); 3) Identity preservation using the ArcFace Distance (Deng et al., 2019).

**Baselines.** We benchmark different state-of-the-art methods, including MagicAnimate (Xu et al., 2024b), MusePose (Tong et al., 2024), MimicMotion (Zhang et al., 2024a), Animate Anyone (Hu, 2024), and ControlNeXt (Peng et al., 2024). To further investigate the effectiveness of our proposed *TalkCuts* dataset, we selected three SOTA methods: MusePose (Tong et al., 2024), Animate Anyone (Hu, 2024), and ControlNeXt (Peng et al., 2024), and fine-tuned them on our dataset.

Table 4: **Quantitative Comparison for pose-guided speech video generation.** Best result is shown in **bold** and the second-best result is shown in underline.

| Method | Video Generation Quality | | | | | ID Preser. |
|---|---|---|---|---|---|---|
| | SSIM↑ | PSNR↑ | LPIPS↓ | FID↓ | FVD↓ | ArcFace Dis. ↓ |
| MagicAnimate Xu et al. (2024b) | 0.731 | 18.397 | 0.235 | 125.500 | 893.230 | 0.552 |
| MimicMotion Zhang et al. (2024a) | 0.759 | 20.572 | 0.168 | 81.820 | 702.410 | 0.435 |
| Animate Anyone Hu (2024) | 0.754 | 20.468 | 0.176 | 93.230 | 789.360 | 0.450 |
| AnimateAnyone Tuned | **0.843** | **24.576** | **0.114** | **57.410** | **456.842** | **0.344** |
| MusePose Tong et al. (2024) | 0.771 | 19.468 | 0.191 | 106.760 | 823.020 | 0.513 |
| MusePose Tuned | 0.785 | 20.933 | 0.164 | 87.000 | 1014.342 | 0.450 |
| ControlNeXt Peng et al. (2024) | 0.746 | 21.584 | 0.149 | 63.150 | 485.118 | 0.409 |
| ControlNeXt Tuned | 0.763 | 21.959 | 0.146 | 62.550 | 480.210 | 0.372 |

**Result Analysis.** The quantitative results are presented in Table 4. After fine-tuning on our proposed *TalkCuts* dataset, all three selected pose-driven models exhibit significant improvements across all key metrics, demonstrating the impact of high-quality training data. Notably, fine-tuned models achieve more natural and detailed body movements, improved facial expression accuracy, and better hand region synthesis under various camera perspectives. Additionally, we observe distinct performance patterns across different models. Animate Anyone (Hu, 2024), a two-stage diffusion model that first learns appearance and subsequently refines motion, retains detailed per-frame appearance well, but suffers from noticeable temporal inconsistency, leading to unstable and jittery motion. On the other hand, ControlNeXt (Peng et al., 2024), which builds upon SVD Blattmann et al. (2023), excels at generating smooth motion, yet struggles with identity consistency, occasionally failing to maintain facial fidelity across frames. While fine-tuning improved facial detail retention, discrepancies between generated faces and reference images persist. Overall, these results highlight that while current pose-guided generation methods benefit greatly from *TalkCuts*, they still face challenges in temporal coherence, identity preservation, and high-fidelity motion synthesis across diverse camera perspectives. These findings further emphasize the importance of developing stronger motion-aware diffusion models for pose-driven human video generation.

## 5 Conclusion

In this paper, we introduced *TalkCuts*, a large-scale benchmark dataset designed to support research in multi-shot speech video generation. *TalkCuts* provides over 500 hours of high-quality speech videos with comprehensive annotations, including transcripts, audio, 2D keypoints, and 3D SMPL-X, covering a wide range of identities and camera shots. To demonstrate the value of this benchmark, we proposed a new generation task and developed *Orator*, a modular pipeline that leverages LLM-based planning and multimodal generation. Through extensive experiments in both pose-guided and audio-driven settings, we show that models trained on TalkCuts consistently outperform existing baselines across multiple aspects, including shot coherence, motion quality, and identity preservation. We hope *TalkCuts* will serve as a foundation for future research in dynamic human video synthesis.

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
