# Supplemental Materials of
# TalkCuts: A Large-Scale Dataset for Multi-Shot Human Speech Video Generation

**Jiaben Chen**[1]* **Zixin Wang**[1]* **Ailing Zeng**[2] **Yang Fu**[1] **Xueyang Yu**[1] **Siyuan Cen**[1]
**Julian Tanke**[3] **Yihang Chen**[4] **Koichi Saito**[3] **Yuki Mitsufuji**[3] **Chuang Gan**[1]
[1]UMass Amherst  [2]Independent Researcher  [3]Sony AI  [4]UC San Diego

## 1 Additional Information on TalkCuts

### 1.1 Manual Screening Process

We outline the detailed steps for ensuring consistency and accuracy in the manual screening process:

1. **Scene Segmentation Validation:** After performing automated scene segmentation using PySceneDetect, human reviewers verify the correctness of the detected shot boundaries. Reviewers ensure that transitions occur at logical points, such as changes in subject focus or significant shifts in speech content. Incorrectly segmented scenes are manually adjusted to improve coherence.

2. **Subject Quality Evaluation:** Each video clip is manually inspected to evaluate the clarity and quality of the human subject within the frame:

- *Clarity:* The subject must be clearly visible without blurring or obstructions.
- *Lighting:* The subject's features must be well-lit and distinguishable.
- *Framing:* The subject must be proportionally centered in the frame.

Clips failing to meet these criteria are discarded.

3. **Consistency and Annotation Accuracy:** Reviewers ensure: 1) *Identity Consistency:* The same individual is consistent across clips for each speaker. 2) *Annotation Validation:* Automated annotations (2D keypoints, 3D SMPL-X, camera trajectories) are verified for a subset of samples. Anomalies are flagged for correction.

4. **General Quality Assessment:** Reviewers ensure: 1) *Speech Alignment:* The subject's lip movements align with the speech audio; 2) *Noise Filtering:* Clips with significant environmental noise or distractions are removed.

5. **Reviewer Training and Quality Audits:** To maintain consistency: 1) Reviewers are trained with examples of acceptable and unacceptable clips. 2) Periodic audits are conducted on random samples to ensure adherence to standards.

This multi-step process ensures that the dataset maintains high visual and audio quality, providing a robust foundation for research.

### 1.2 Data Statistics on Shot Types

Below are the results and corresponding analysis of the total number of clips and the distribution of shot sizes for each identity.

Shown in Fig. 1, the bar chart in the left illustrates the frequency distribution of clip counts across predefined ranges. The X-axis represents different ranges of clip counts (0-5, 5-12, 12-30, 30-70,

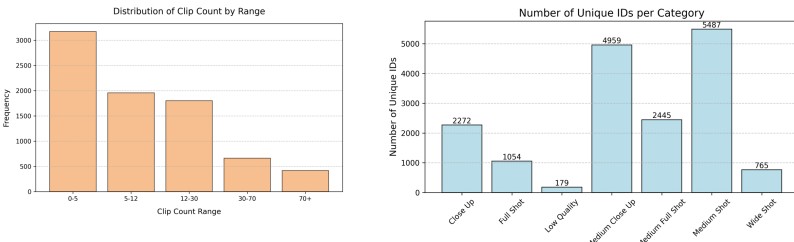

Figure 1: **Statistics of dataset clips: Left** - Clip Count Distribution per ID grouped by range. **Right** - Distribution of Unique IDs across Shot Categories.

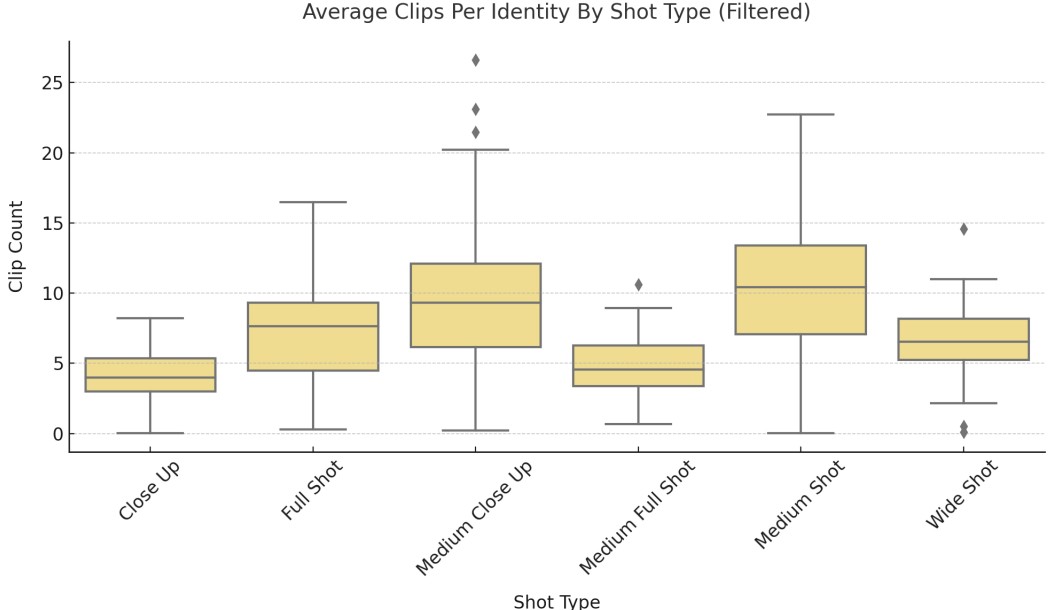

Figure 2: **Distribution of Average Clip Counts Per Identity Across Shot Types:** the boxplot shows the distribution of clip counts per identity across six shot types. The Y-axis represents the clip counts, and the X-axis categorizes the shot types. Each box represents the interquartile range (IQR), with the median as a horizontal line inside, whiskers indicating variability, and outliers shown as points.

and 70+), while the Y-axis indicates the frequency, i.e., the frequency statistic represents the number of distinct clips associated with each ID across the entire dataset, falling within specific ranges. The Y-axis value corresponds to the number of IDs in each range. The bar chart on the right of Fig. 1 visualizes the number of unique IDs (identities) associated with each shot category. The X-axis represents the shot categories, including Close Up, Full Shot, Low Quality, Medium Close Up, Medium Full Shot, Medium Shot, and Wide Shot. The Y-axis shows the count of unique IDs for each category.

Additionally, shown in Fig. 2, each box represents the interquartile range (IQR), with the median as a horizontal line inside, whiskers indicating variability, and outliers shown as points. Medium Shot and Medium Close Up dominate with higher medians and broader distributions, while Full Shot and Wide Shot have lower medians and fewer outliers. This visualization highlights the variability and prevalence of shot types across identities.

## 2  Additional Evaluation

### 2.1  Human Evaluation for Camera Shot Changes directed by LLM

For subjective evaluation of camera shot changes generated by the LLM, we conduct an experiment with 20 participants, each rating several criteria on a 1-5 scale (1 = poor, 5 = excellent). We compare our model, ground truth (GT), a rule-based system (shots based on speech length, punctuation, and keywords), zero-shot LLM, and a random baseline (shots randomly assigned). Evaluators will assess from the following aspects:

- Shot Coherence: measures the logical flow between camera shots and evaluates how well the transitions follow the speech content. Evaluators will assess whether the changes in shots are smooth and whether the cuts feel appropriate based on the context. For instance, sharp and abrupt cuts during calm moments would detract from coherence, while fluid transitions during significant speech segments should enhance it.

- Visual Engagement: aimed at evaluating whether the video remains visually captivating and holds the viewer's attention throughout.

- Shot-Type Appropriateness: refers to how suitable the selected shot types (e.g., close-up, medium shot, wide shot) are in relation to the content being delivered. Evaluators will consider whether emotional intensity or important speech moments are reflected with close-up shots and whether wider shots are used to contextualize broader topics or transitions.

- Overall Quality: provides a holistic evaluation of the video, capturing the combined effectiveness of shot selection, transitions, and flow.

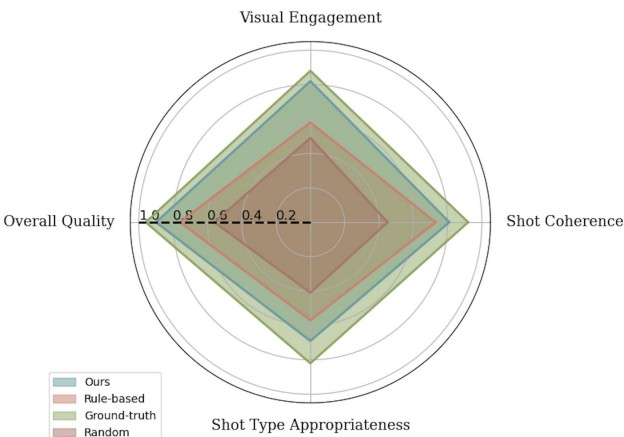

Figure 3: **User study** on camera shot changes directed by LLM.

The results of the user study are presented in Fig. 3. As demonstrated, our DirectorLLM consistently outperforms the rule-based system, zero-shot LLM, and random baselines in all evaluation criteria. While the ground truth still holds the highest ratings, our model closely approaches its performance, indicating the effectiveness of LLM-driven shot changes and the smoothness of transitions generated by our approach.

## 3  Details of Retrieval-augmented Generation

### 3.1  Retrieval-augmented Generation Process

We provide a detailed illustration of the RAG process in Fig. 4. We aim to enhance the shot transition performance of LLMs using RAG. To achieve this, we use scripts with annotated shot transitions from the training dataset as the RAG corpus. The training dataset, composed of text scripts, is converted into an embedding dataset using the OpenAI text-embedding-small model.

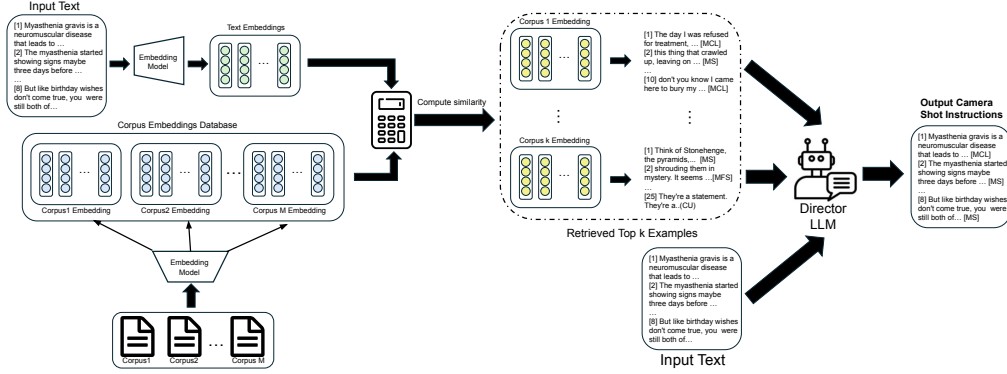

Figure 4: **Pipeline of RAG** in detail.

For each input script, we similarly convert it into a text embedding using the same embedding model. Then, utilizing the FAISS (Douze et al., 2024) tool, we calculate the L2 distance between the input text embedding and each embedding in the embedding dataset. The top 5 files with the smallest distances are selected as the context, which is provided alongside the input script as input to the model.

## 3.2 Retrieval Examples

As shown in Figure 5, these are two examples of RAG assisting LLM in making shot transitions. In the first example, the **bold blue** portions of the script and relevant documents both express the love between a boy and a girl in a poetic manner. In the second example, the **bold blue** text highlight the importance of intimate relationships in helping humans confront pain and illness. The shot transition results in both examples align with those obtained through our RAG-based approach. The two examples respectively illustrate that the documents retrieved by RAG share similarities with our input scripts in terms of content or narrative logic. This demonstrates that the documents retrieved by RAG can indeed assist the LLM in making shot transitions.

# 4 Additional Related Works

## 4.1 Human Video Datasets.

Recently, various datasets derived from public platforms such as TikTok and YouTube have been introduced to advance human video generation research. For example, the TikTok dataset (Jafarian and Park, 2021) includes 340 short video clips, each lasting 10-15 seconds, primarily featuring dancing humans, while UBC-Fashion (Zablotskaia et al., 2019) consists of 500 fashion-related clips. However, these datasets are limited in both scale and quality. To overcome these limitations, several synthetic datasets (Varol et al., 2017; Patel et al., 2021; Cai et al., 2021; Yang et al., 2023) have been developed, significantly enhancing the diversity of backgrounds and the scale of training data. For instance, Bedlam (Black et al., 2023) includes thousands of clips with over 1.5 million frames, featuring high-resolution rendered humans in realistic environments.

Recognizing the growing importance of multi-modal data for training, recent datasets Li et al. (2021); Siarohin et al. (2021); Luo et al. (2020) have incorporated various modalities. Furthermore, advancements in annotation tools have facilitated the creation of large-scale, highly realistic datasets.HumanVid (Wang et al., 2024) consist of more than 50M frames and these frames are well annotated and BEAT2 has more than 32M frames with a high resolution of 1080P. However, these datasets are still limited to identity numbers, which may constrain the ability of generalizations. While datasets like MENTOR (Corona et al., 2024) exists that have over 80k identities dynamic gestures, the dataset remains private. To the best of our knowledge, we are the first public human video datasets that contains thousands of identities.

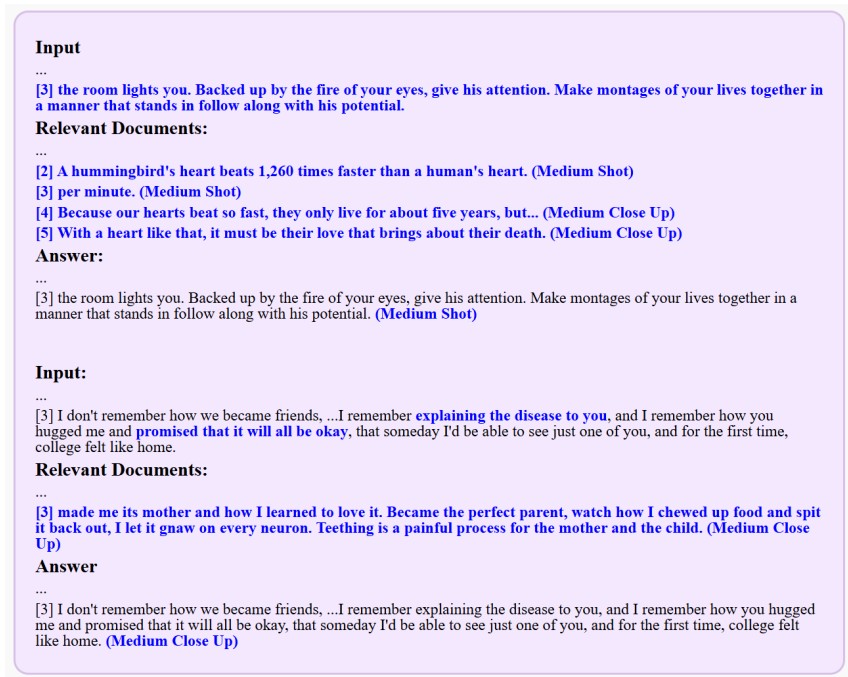

Figure 5: **Examples of RAG-Assisted Shot Transitions**:The **bold blue** text highlights similarities between the input script and the content retrieved from RAG documents.

## 4.2 Audio-Driven Human Video and Motion Generation

Holistic body motion generation from speech involves synthesizing whole-body motions (Li et al., 2021; Qi et al., 2024). Recognizing that audio signals convey more than just semantic content, (Yi et al., 2023) propose a method to generate holistic body movements by segmenting the audio signal into different components, each guiding a separate motion generation process. Similarly, learning from masked gesture data, EMAGE (Liu et al., 2023) utilizes four compositional VQ-VAEs for generation. Witnessing the success of diffusion models, more and more works (Chen et al., 2024) began to utilize a diffusion-based structure. MotionCraft (Bian et al., 2024) exemplifies this trend, using a unified DiT structure to incorporate multimodal controls and achieving state-of-the-art results in audio-to-motion generation. In the domain of audio-driven video generation, preliminary works (Sun et al., 2023; Tian et al., 2024; Ji et al., 2024) have primarily concentrated on facial regions, ensuring a high degree of consistency between lip movements and the semantic content of the corresponding audio. To expand the generated region, (Corona et al., 2024) synthesizes half-body human videos, while Make-Your-Anchor (Huang et al., 2024) generates anchor-style full-body videos by translating audio into detailed torso and hand movements using a two-stage diffusion model. ANGIE (Liu et al., 2022) employs an unsupervised feature to model body motion while DiffTED (Hogue et al., 2024) decouples motion from gesture videos while preserving additional appearance information.

## 4.3 Movie & Cartoon Understanding and Generation

Recent advancements in generative video models have integrated autoregressive frameworks, diffusion models, and large language models (LLMs) to address challenges in long-form, multimodal video generation and animation. Early methods such as StoryGAN (Li et al., 2019) and PororoGAN(Zeng et al., 2019) used GAN-based models for visual storytelling but were limited by contextual inconsistencies in generated frames. To address these limitations, Anim-Director (Li et al., 2024) uses LLMs to autonomously manage the entire animation creation process, refining narratives, generating scripts, and producing contextually coherent animations from brief inputs. Similarly, MovieDreamer (Zhao et al., 2024) combines autoregressive models with diffusion rendering to maintain narrative and character consistency in long-form videos, decomposing complex stories into manageable segments for high-quality visual synthesis. Recent research has also explored using LLMs as "directors" in

video generation, where they coordinate various elements similar to a human director managing a film production. (Zhu et al., 2024) use LLMs to decompose complex prompts into sub-tasks, enabling precise control over 3D scene generation.(Argaw et al., 2022) introduce a benchmark for AI-assisted video editing, focusing on decomposing movie scenes into individual shots based on attributes like camera angles and shot types. This structured representation of shots is conducive to LLM-based systems, which can then manage and edit sequences in a manner similar to a human editor, enhancing the automation of video editing tasks. Additionally, (Rao et al., 2020) propose a subject-centric model to classify shot types, which can enhance LLM-guided video generation by providing structured visual cues. This research suggests that LLMs are well-suited for directing complex video creation processes.

## 5 Limitations and Future Works.

While TalkCuts provides a comprehensive benchmark for multi-shot speech video generation, several limitations remain. First, the dataset primarily features speakers in static environments, with limited interaction with props or stage movement (e.g., using a microphone or walking). Expanding the dataset to include dynamic speaker-object interactions could better support embodied generation tasks. Second, although TalkCuts captures varied camera shots, it lacks audience engagement cues (e.g., gaze shifts, eye contact, facial reactions), which are essential in real-world speech scenarios like talk shows or interviews. Additionally, while the proposed system handles multi-shot transitions effectively, it does not yet incorporate moving camera dynamics, which would further enhance the realism of the generated videos. As future work, we aim to explore moving camera integration leveraging advanced camera control modules.

## 6 Potential Practical Application

The practical value of multi-shot speech video generation lies in its potential to revolutionize content creation across various industries by automating a traditionally labor-intensive and creative process. Key applications include:

- Entertainment and Media Production: This technology enables the efficient creation of dynamic, multi-shot speech videos for films, TV shows, and online content. By automating camera transitions, gesture synthesis, and vocal delivery, our system reduces the need for extensive manual editing and enhances the storytelling quality.

- Education: Multi-shot speech videos can be used to create engaging educational content, such as lectures or tutorials, where dynamic camera angles and gestures help maintain viewer interest and improve the conveyance of information.

- Corporate Communications: Businesses can use this technology to generate polished speech videos for presentations, product launches, or training sessions, offering a cost-effective way to produce professional-quality content.

- Content Creation for Social Media: Influencers and creators can leverage multi-shot speech video generation to produce compelling, visually engaging videos for platforms like YouTube, TikTok, or Instagram without requiring advanced editing skills or significant production resources.

- Virtual and Augmented Reality: Multi-shot speech videos could serve as a foundational component for immersive virtual presentations or augmented reality experiences, where dynamic and lifelike speech scenarios are crucial.

By addressing the complex challenge of generating long-form speech videos with dynamic camera shots, our work provides a foundation for these applications. The integration of the DirectorLLM with multimodal generation modules demonstrates a novel approach to orchestrating speech, motion, and visual elements in a cohesive manner. Our system reduces the barriers to high-quality video production, enabling creativity and innovation across industries. It offers a scalable solution that can adapt to various content requirements while maintaining consistency and realism. We believe that our research not only advances the technical capabilities in this domain but also opens up new possibilities for practical applications that can have a positive impact on entertainment, education, business, and more.

# 7 Potential Risks

Our proposed method presents risks related to potential misuse for misinformation campaigns and large-scale generation of fake news. To mitigate these concerns, we have carefully curated the dataset to include only innocuous topics such as education, entertainment, and public speaking in neutral settings. By focusing on benign subjects, we aim to minimize the potential for our work to be exploited for malicious purposes, while still demonstrating the effectiveness of our approach in a controlled and ethical manner. We are committed to responsible research practices and have taken deliberate steps to ensure that our contributions do not inadvertently contribute to the spread of misinformation or harmful content. Additionally, we encourage further exploration of ethical safeguards and detection mechanisms to prevent misuse.