# OpenReview forum: "TalkCuts: A Large-Scale Dataset for Multi-Shot Human Speech Video Generation"
_NeurIPS.cc/2025/Datasets_and_Benchmarks_Track — NeurIPS 2025 Datasets and Benchmarks Track poster_

### Official Review · Reviewer_NpwC · 2025-06-28

**Rating:** 5
**Confidence:** 4

**Summary:**

This paper proposed a large-scale benchmark dataset designed to support research in multi-shot speech video generation. Correspondingly, a new method has been proposed for generating multi-shot human body videos. Overall, this dataset is meaningful. Currently, open-source human body datasets have limitations in terms of the number of videos and the diversity of shots.

**Dataset Code Accessibility:**

Yes

**Ethical Considerations:**

No, there are no or only very minor ethics concerns

**Final Justification:**

My concerns have been addressed.

**Limitations Weaknesses:**

1. There are still obvious artifacts in the generated video. There are obvious pixel changes in some frames, and there is also jitter between the shots.
2. Figure 2 and Figure 3 are to some extent repetitive. The process of dataset collection and annotation should be added.
3. Are the clips between consecutive shots in a video continuous?
4. Will the proposed baseline code be open source?

**Strengths Contributions:**

1. Compared with the existing open-source datasets, the proposed dataset has an advantage in terms of scale and defines and classifies different shots.
2. The generation task of multi-shot human body videos is meaningful.

---

> ### Author Rebuttal · Authors · 2025-07-31
>
> Dear Reviewer,
>
> We thank the reviewer for the positive reviews and insightful suggestions.
>
> > There are still obvious artifacts in the generated video. There are obvious pixel changes in some frames, and there is also jitter between the shots.
>
> Thank you for pointing out the visual artifacts and temporal jitter observed in some of the generated frames. We acknowledge that temporal inconsistency, such as flickering and jitter between frames or across camera shots, remains a key challenge in video generation, especially in image animation-based pipelines. This limitation is not unique to our method but is a well-known problem in the broader video generation literature. The task of multi-shot video generation is highly challenging and that we provide only a first baseline approach, as our main contribution is the novel dataset.
>
> Also, we acknowledge that our current model does not explicitly address fine-grained details such as hand articulation. As our framework focuses on establishing a unified multi-shot video generation baseline, we have not yet introduced component-level refinements. Even state-of-the-art methods like Hallo3 [1] exhibit similar issues, as can be observed in their official GitHub results (the little girl in front of the fire example), where hand artifacts are still visible. A promising future direction is to integrate recent advances such as CyberHost [2], which introduces region-level attention and a hand clarity scoring module to enhance the fidelity of articulated regions like hands. Such techniques could be particularly beneficial in improving detail preservation and reducing perceptual artifacts across shots.
>
> While our method already demonstrates clear improvements in identity preservation and human details over prior baselines by leveraging the TalkCuts dataset, we recognize that the above challenges remain open. We plan to investigate these directions in future work, particularly by exploring methods that improve both local visual quality and global temporal stability in multi-shot scenarios.
>
> [1] Cui, Jiahao, et al. "Hallo3: Highly dynamic and realistic portrait image animation with video diffusion transformer." Proceedings of the Computer Vision and Pattern Recognition Conference. 2025.
>
> [2] Lin, Gaojie, et al. "Cyberhost: A one-stage diffusion framework for audio-driven talking body generation." The Thirteenth International Conference on Learning Representations. 2025.
>
>
>
> > Figure 2 and Figure 3 are to some extent repetitive. The process of dataset collection and annotation should be added.
>
> Thank you for the thoughtful comments! We agree that Figure 2 and Figure 3 exhibit some redundancy. Our original intention was to use Figure 2 as a teaser image to highlight the novel task of multi-shot speech video generation. This figure is designed to give an overview of our proposed system, which takes as input text lines and reference images, and uses a Director LLM to direct camera transitions, speaker gestures, and audio instructions, ultimately producing coherent multi-shot speech videos. Figure 3, in contrast, was intended to illustrate the detailed pipeline of our proposed baseline system, emphasizing the integration of retrieval-augmented generation (RAG) for planning and the use of a video diffusion model for generation. In the future version, we will merge Figure 2 and Figure 3 to avoid redundancy while retaining the key information from both figures.
>
> We also fully agree with the second suggestion that a visual illustration of the dataset collection and annotation process would significantly improve clarity. Currently, this process is only described in text. We plan to add a new figure to clearly visualize the dataset pipeline, including data sourcing, speech segmentation, camera shot annotation, multimodal annotation (e.g., keypoints, SMPL-X), and quality filtering. This new figure will replace one of the existing figures, thereby enhancing the overall presentation while maintaining visual balance in the paper.
>
>
> > Are the clips between consecutive shots in a video continuous?
>
> Thank you for the question. In our original data collection, the raw speech videos were composed of fully continuous clips—each consisting of an uninterrupted segment of a talk or presentation. Thus, the initial clips between consecutive shots were indeed continuous in content.
>
> However, during our data filtering and manual screening process, certain segments were intentionally removed to ensure the quality and relevance of the dataset. Specifically, we excluded clips that met any of the following conditions:
>
> - The speaker was not visible (e.g., the camera was focused on the audience).
>
> - The shot did not include the speaker but instead showed irrelevant background or scene transitions.
>
> - The clip had poor visual quality (e.g.m heavy motion blur).
>
> As a result, while the majority of the video segments remain continuous, there are occasional gaps in the final dataset where such clips were removed. Importantly, we ensured that each individual clip is temporally continuous within itself, preserving natural motion and speech flow. We believe this trade-off was necessary to guarantee the overall quality and consistency of the dataset, especially for training generative models that rely on clean and relevant speech-video alignment.
>
> > Will the proposed baseline code be open source?
>
> Yes, we plan to open source both the dataset and the full implementation of our proposed baseline.
>
>
> We sincerely appreciate your time for reviewing this paper and raising the valuable suggestions! If you have any further questions, feel free to let us know and we'll be happy to address them.
>
> Best,
>
> Authors

---

> > ### Comment · Reviewer_NpwC · 2025-08-04
> >
> > Thank you for the author's reply. My concerns have been addressed, and I will keep my positive score.

---

### Official Review · Reviewer_BTRq · 2025-06-29

**Rating:** 4
**Confidence:** 4

**Summary:**

This paper introduces a novel task: speech-driven video generation with dynamic camera shots. To support this task, the authors present TalkCuts, a large-scale dataset specifically designed to capture multi-shot video dynamics aligned with long-form speech. Furthermore, the paper proposes Orator, an agent-wise framework built around a central module, DirectorLLM, to effectively model this complex generation process. Experimental results across multiple tasks demonstrate the effectiveness and generalizability of the proposed approach and dataset.

**Dataset Code Accessibility:**

Partly

**Dataset Code Comments:**

Only a few samples of dataset are available. I do not see any code.

**Ethical Considerations:**

No, there are no or only very minor ethics concerns

**Final Justification:**

My concerns were well addressed.

**Limitations Weaknesses:**

1. The use of three separate reference image types (head, half-body, full-body) is questionable. Given a full-body image, the model should be able to infer head and half-body views.
2. The paper lacks comparisons with camera motion–based methods. Camera-controlled generation (e.g., smooth zoom-ins) appears more practical than the proposed setup. Moreover, the dataset lacks explicit camera motion annotations.
3. Section 4.1 should include comparisons with camera-controlled video generation methods.
4. The video results are underwhelming, especially when compared to recent approaches like OmniHuman. Hand generation, in particular, is noticeably poor. This raises concern about the quality of the dataset since Hallo3 shows good hand generation results. However, only a few samples of dataset are attached to the provided link.

**Strengths Contributions:**

1. The presentation is clear.
2. The task is novel, and the proposed dataset is meaningful to the community.
3. The experimental analysis is sufficient.

---

> ### Author Rebuttal · Authors · 2025-07-31
>
> Dear Reviewer,
>
> We thank the reviewer for the detailed reviews and insightful suggestions.
>
> > The use of three separate reference image types is questionable ...
>
> Thank you for the thoughtful question. We acknowledge that our current pipeline requires a reference image for each shot. The task of multi-shot video generation is highly challenging and we provide only a first simple yet effective baseline approach, as our main contribution is the novel dataset.
>
> We agree that an ideal system would take a single reference image of the speaker and automatically synthesize novel camera shots, such as transitioning from a close-up to a wide shot. Regarding the generation of novel-view video clips, this represents an exciting and challenging direction for future research. While there have been advancements in novel-view synthesis for objects [1,2], the ability to generate high-quality, dynamic novel views of humans, such as transitioning from a half-body shot to a full-body view, remains an unsolved problem. Current methods often struggle to maintain fidelity to both the subject's identity and the background across significantly different viewpoints.
>
> However, our proposed TalkCuts dataset provides a unique opportunity to explore this problem. Since the dataset includes many videos of the same identity captured from multiple viewpoints, it could be leveraged to develop methods that tackle novel-view synthesis for dynamic human subjects. We agree that this is a promising direction, and we plan to investigate it further in future work. Thank you again for suggesting this exciting avenue!
>
> [1] Sargent, Kyle, et al. "Zeronvs: Zero-shot 360-degree view synthesis from a single real image." arXiv preprint arXiv:2310.17994 (2023).
>
> [2] Van Hoorick, Basile, et al. "Generative camera dolly: Extreme monocular dynamic novel view synthesis." European Conference on Computer Vision. Springer, Cham, 2025.
>
> > 1)The paper lacks comparisons with camera-controlled methods... 2) Moreover, the dataset lacks explicit camera motion annotations...
>
> Thank you for raising these important points. We address them below:
>
> **1. Comparison with camera-controlled methods:**
> We acknowledge that camera-controlled generation is an important line of work [4,5,6]. However, we emphasize that camera motion and shot transitions are two distinct techniques in filmmaking, serving different narrative and aesthetic functions:
> - Camera motion is commonly used in films and games, where the camera moves around a subject, often directed by a predefined trajectory.
> - In contrast, camera shot switching is widely adopted in speech scenarios, such as TED talks and talk shows, where sudden transitions between fixed-view shots are used to maintain viewer engagement during long monologues.
>
> In our work, we focus on the latter, which has been largely overlooked in prior video generation research, due to the challenging nature of the task and the lack of existing datasets. In this work we attempt to close this gap by (1) providing for the first time a multi-shot talking human video dataset, and (2) provide a simple baseline approach.
>
> A direct comparison with camera-controlled methods is non-trivial, as prior works like [4,5] rely on explicit camera trajectory inputs to control movement. In contrast, our shot transitions are content-driven, determined from input scripts, and involve discrete view changes, where no continuous trajectory exists.
>
> That said, we agree that incorporating camera motion can further enhance realism, especially for the movie domain. Our pipeline is modular and flexible, it can incorporate camera motion modules. For instance, similar to [4], we plan to add an additional camera control branch into the diffusion process [5] to condition on camera trajectories, or we could adopt a camera embedding module [6] to enable controllable cinematic effects like zooming from text.
>
>
> **2. Camera motion annotations in the dataset:**
> While most of our dataset focuses on static shots with cut transitions, we also include a subset (~10%) of speech videos that contain camera motion. For these, we reconstruct camera trajectories using TRAM [3]. These trajectories have been explicitly annotated and will be released as part of our dataset, providing a foundation for future research on camera-motion-aware video generation.
>
> [3] Wang, Yufu, et al. "TRAM: Global Trajectory and Motion of 3D Humans from in-the-wild Videos." European Conference on Computer Vision. Cham: Springer Nature Switzerland, 2024.
>
> [4] Wang, Zhenzhi, et al. "Humanvid: Demystifying training data for camera-controllable human image animation." Advances in Neural Information Processing Systems 37 (2024): 20111-20131.
>
> [5] Bai, Jianhong, et al. "Recammaster: Camera-controlled generative rendering from a single video." arXiv preprint arXiv:2503.11647 (2025).
>
> [6] Yang, Shiyuan, et al. "Direct-a-video: Customized video generation with user-directed camera movement and object motion." ACM SIGGRAPH 2024 Conference Papers. 2024.
>
> > The video results are underwhelming...
>
> Thank you for the detailed feedback. We address each point below:
>
> **1. Hand generation quality**
>
> We acknowledge that our current baseline does not explicitly address fine-grained regions such as hands. Multi-shot video generation is inherently challenging, and our paper presents the first baseline for multi-shot speech-driven video generation, without introducing component-level refinements.
>
> That said, even state-of-the-art methods such as Hallo3 can exhibit hand artifacts. For instance, in the little girl in front of the fire example on their official GitHub page, hand distortions are still clearly visible. A promising direction for future improvement is to integrate techniques like CyberHost [7], which use hand clarity scores and region-level attention to enhance fidelity in articulated regions. We believe such modules can greatly improve hand detail and consistency across shots.
>
> **2. Comparison with OmniHuman**
>
> We agree that OmniHuman produces impressive results. However, it is important to note key differences: 1) OmniHuman is trained on a private 18.7K-hour multi-modal dataset, which is orders of magnitude larger than any current public dataset; 2) It uses mixed conditioning on audio and pose during training, which differs from our setup.
>
> In contrast, our work’s primary contribution is the first public, large-scale dataset (500+ hours) focused specifically on speech-driven, multi-shot human video generation, which is a domain that is under explored in prior work.
>
> **3. Dataset quality and model strength**
>
> We respectfully clarify that the model quality is not indicative of dataset quality. In our submission, we used CogVideoX as the backbone. This backbone is relatively weak, as previously noted, Hallo3 (using CogVideoX) also exhibits similar artifacts when trained on the same base.
>
> To address this, we recently rebuilt our pipeline using the newly released Wan 2.1 model [8]. The method remains aligned with the approach described in our paper, and is trained on our dataset using a cross-attention mechanism for audio conditioning, similar to [9]. The newly generated videos exhibit much better detail and significantly improved temporal consistency, with noticeably reduced flickering artifacts. Although the rebuttal policy prevents us from sharing video demos, we conducted evaluations on our test set. The updated model shows significant improvements in visual detail, temporal consistency, and lip-sync alignment, as the table below:
>
> | **Method**   | FID ↓     | FVD ↓     | PSNR ↑   | SSIM ↑   | LPIPS ↓  | Sync-C ↑ | Sync-D ↓ | Sub. Cons. ↑ | Back. Cons. ↑ | Dynamic Deg. ↑ | Motion Smo. ↑ |
> |---------------|--------|--------|--------|--------|---------|--------|--------|--------|--------|--------|-----------|
> | SadTalker   | 159.78    | 926.01    | 12.46    | 0.356    | 0.572    | 4.57 (2nd)   | 8.89 | 91.72%         | **98.99%**      | 19.42%             | **99.81%**           |
> | EchoMimicV2 | 177.22    | 1770.22   | 14.88    | 0.582    | 0.511    | 1.94       | 9.65       | 88.61%         | 93.25%          | **87.10%**          | 99.30%              |
> | Hallo3      | 57.28 | 855.84 (2nd)| 19.22 | 0.644 | 0.215 (2nd)| 3.84       | 9.76       | 95.81%   | 94.33%          | 54.84%             | 99.45%              |
> | Ours w. Cog    | 45.86 (2nd) | **622.76** | 20.61 (2nd) | 0.708 (2nd) | 0.198 | 4.35  | 8.32 (2nd)| 96.24%  (2nd)  | 95.42%   | 68.48%       | 99.63%        |
> | Ours w. Wan| **43.75** | 694.23 (2nd) | **21.02** | **0.715** | **0.198**| **4.82**  | **8.12**   | **96.72%**     | 96.01% (2nd)   | 72.34% (2nd)   | 99.74% (2nd)            |
>
> **4. Dataset samples and quality assurance**
>
> Due to Kaggle’s file size limit, we provided a random sample of 25 videos. However, we assure the reviewer that this sample is representative. Furthermore, our demo video on the dataset overview page includes nearly 100 clips from the dataset to showcase diversity and quality.
>
> To ensure data quality, we conducted a multi-step manual screening process (described in Section 2.1 of the supplemental), which includes validation of scene segmentation, subject visibility, annotation correctness, and lip-sync alignment. This gives us high confidence in the quality and research value of the dataset.
>
> [7] Lin, Gaojie, et al. "Cyberhost: A one-stage diffusion framework for audio-driven talking body generation." The Thirteenth International Conference on Learning Representations. 2025.
>
> [8] Wan, Team, et al. "Wan: Open and advanced large-scale video generative models." arXiv preprint arXiv:2503.20314 (2025).
>
> [9] Kong, Zhe, et al. "Let Them Talk: Audio-Driven Multi-Person Conversational Video Generation." arXiv preprint arXiv:2505.22647 (2025).
>
> We sincerely appreciate your time for reviewing this paper and raising the valuable suggestions! If you have any further questions, feel free to let us know and we'll be happy to address them.
>
> Best,
>
> Authors

---

> ### Comment · Area_Chair_7aWK · 2025-08-08
>
> Dear Reviewer BTRq,
>
> Since the Discussion period will end soon, could you please review the authors' rebuttal and provide your comments? Thank you!
>
> Best,
>
> AC

---

### Official Review · Reviewer_bXnd · 2025-06-30

**Rating:** 4
**Confidence:** 3

**Summary:**

This paper proposes a large-scale dataset named TalkCuts for multi-shot human speech video generation, offering 164K clips (500+ hours of 1080P video) with diverse camera shots (close-up, full-body), 2D keypoints, 3D SMPL-X annotations, and textual descriptions across 10K+ identities. The authors also construct a baseline to showcase the value of this dataset, the presented Orator is acted as a multi-faceted director, orchestrating detailed specifications for camera transitions, speaker gesticulations, and vocal modulation. Experiments show TalkCuts enhances cinematographic coherence and visual quality compared to baselines in pose-guided and audio-driven settings.

**Dataset Code Accessibility:**

Yes

**Ethical Considerations:**

No, there are no or only very minor ethics concerns

**Final Justification:**

Thank you for the author's response, My concerns were well addressed and I maintain my positive score unchanged.

**Limitations Weaknesses:**

1.	As in Line142, how to ensure that the refined hand pose by HaMeR maintains consistent wrist poses with SMPLer-X?
2.	Combining the pipeline and the final generated videos, it seems that there has been a camera shot switch at each reference image. Does the input reference images of Orator serve as keyframes?
3.	As described in Line248, the final video was obtained through concatenated different clips. For a unified Video Diffusion Model, is the length of each clip consistent?
4.	From the manuscript, we know that the relevant annotations in the dataset are obtained using existing methods. Is it necessary to conduct metric evaluating on the dataset itself.
5.	Key elements of Orator seem to be incremental. Considering the amount of work involved in TalkCuts and the experimental results presented by the authors, this is not a key factor affecting my score.

**Strengths Contributions:**

1.	With 164K clips and 500+ hours of 1080P video, TalkCuts outperforms prior datasets (e.g., TikTok, TED Talks) in scale, covering diverse camera shots, speaker identities, and speech scenarios (TED talks, stand-up comedy). The multi-modal annotations (2D keypoints, 3D SMPL-X) enable comprehensive multimodal learning. Importantly, compared to single-shot videos, adopting this dataset helps to enhance the cinematographic coherence and visual appeal of generated multi-shot speech videos.
2.	The proposed baseline is meaningful, in which LLM can serve as a multi-faced director, generating context-aware instructions for camera transitions, gestures, and vocal delivery. This modular design integrates seamlessly with diffusion models, demonstrating coherent long-form video synthesis.
3.	Quantitative results (e.g., FID, PSNR, Sync-C) show significant improvements over state-of-the-art baselines in video quality, lip sync, and motion smoothness. Qualitative comparisons highlight fewer artifacts (facial distortions, motion blur).

---

> ### Author Rebuttal · Authors · 2025-07-31
>
> Dear Reviewer,
>
> We thank the reviewer for the detailed reviews and insightful suggestions.
>
> > How to ensure that the refined hand pose by HaMeR maintains consistent wrist poses with SMPLer-X
>
> Thank you for the question. We follow the integration strategy discussed in the official HaMeR [1] GitHub repository. Specifically, we calculate the global rotations of the elbow and the entire hand, and then use them to compute the local wrist pose. The finger poses can be transferred directly from HaMeR’s MANO prediction into the SMPL-X hand joints. This ensures the refined hand pose maintains kinematic consistency with the SMPLer-X body.
>
> [1] Pavlakos, Georgios, et al. "Reconstructing hands in 3d with transformers." Proceedings of the IEEE/CVF Conference on Computer Vision and Pattern Recognition. 2024.
>
> > Combining the pipeline and the final generated videos, it seems that there has been a camera shot switch at each reference image. Does the input reference images of Orator serve as keyframes?
>
> Thank you for the thoughtful question. Yes, our current approach requires a reference image for each specified camera shot. Each clip is generated independently, and the corresponding reference image serves as a keyframe, guiding the appearance and identity under the given camera view. This design ensures consistency in facial details and body appearance across different shot types.
>
> We agree that an ideal system would take a single reference image of the speaker and automatically synthesize novel camera shots, such as transitioning from a close-up to a full-body or wide shot. This represents an exciting and challenging future direction. Regarding the generation of novel-view clips, this represents an exciting and challenging direction for future research. While there have been advancements in novel-view synthesis for objects (e.g., [2,3]), the ability to generate high-quality, dynamic novel views of humans, such as transitioning from a half-body shot to a full-body view, remains an unsolved problem. Current methods often struggle to maintain fidelity to both the subject's identity and the background across significantly different viewpoints
>
> However, our proposed TalkCuts dataset provides a unique opportunity to explore this problem. Since the dataset includes many videos of the same identity captured from multiple viewpoints, it could be leveraged to develop methods that tackle novel-view synthesis for dynamic human subjects. We agree that this is a promising direction, and we plan to investigate it further in future work. Thank you again for suggesting this exciting avenue!
>
> [2] Sargent, Kyle, et al. "Zeronvs: Zero-shot 360-degree view synthesis from a single real image." arXiv preprint arXiv:2310.17994 (2023).
>
> [3] Van Hoorick, Basile, et al. "Generative camera dolly: Extreme monocular dynamic novel view synthesis." European Conference on Computer Vision. Springer, Cham, 2025.
>
> > For a unified Video Diffusion Model, is the length of each clip consistent?
>
> Thank you for the question. During training, we use fixed-length clips to ensure stable and efficient learning. However, during inference, each clip’s length is dynamically determined based on the duration of the corresponding input audio segment. The model is capable of handling variable-length inputs at inference time. The final video is constructed by concatenating these variable-length clips.
>
>
> > From the manuscript, we know that the relevant annotations in the dataset are obtained using existing methods. Is it necessary to conduct metric evaluating on the dataset itself.
>
> Thank you for the thoughtful question. Our dataset includes several types of annotations, all obtained using state-of-the-art methods:
>
> - 2D keypoints: Annotated using DWPOSE [9], a top-performing whole-body pose estimator.
> - Text descriptions: Generated using the SOTA vision-language model Qwen-VL 2.5 [10], which produces accurate summaries based on video content.
> - SMPL-X parameters: Following the best strategy from MotionX++ [8], we combine SMPLer-X [4] for body, HaMeR [1] for hands, and EMOCA [7] for FLAME-based facial expression recovery.
>
> To assess the quality of the SMPL-X annotations, we conducted a perceptual study inspired by SHOW [5], where users evaluated whether the reconstructed face, hands+body, and full-body poses matched the input video across 30 clips. Results in the below table show our final annotation strategy consistently outperforms alternative pipelines.
>
> | Method        | face  | body and hands | holistic body |
> |---------------|-------|-------|----------------|
> | SMPLer-X [4] | 0.133 | 0.633 | 0.667  |
> | SHOW [5]  | **0.867** | 0.600 | 0.633 |
> | OSX [6] + EMOCA [7] | 0.833 |  0.567 | 0.600 |
> | **SMPLer-X [4] + EMOCA [7] + HaMeR [1] (Ours)** | 0.833 | **0.700** | **0.733** |
>
> In addition, we applied a multi-step manual screening process (described in Section 2.1 of the supplemental), which includes validation of scene segmentation, subject visibility, annotation correctness, and lip-sync alignment. Reviewers were trained, and periodic audits ensured annotation and content quality.
>
> Together, this combination of SOTA methods, human verification, and perceptual validation provides confidence in the quality of our annotations.
>
> [4] Cai, Zhongang, et al. "Smpler-x: Scaling up expressive human pose and shape estimation." Advances in Neural Information Processing Systems 36 (2023): 11454-11468.
>
> [5] Yi, Hongwei, et al. "Generating holistic 3d human motion from speech." Proceedings of the IEEE/CVF Conference on Computer Vision and Pattern Recognition. 2023.
>
> [6] Lin, Jing, et al. "One-stage 3d whole-body mesh recovery with component aware transformer." Proceedings of the IEEE/CVF Conference on Computer Vision and Pattern Recognition. 2023.
>
> [7] Daněček, Radek, Michael J. Black, and Timo Bolkart. "Emoca: Emotion driven monocular face capture and animation." Proceedings of the IEEE/CVF Conference on Computer Vision and Pattern Recognition. 2022.
>
> [8] Zhang, Yuhong, et al. "Motion-x++: A large-scale multimodal 3d whole-body human motion dataset." arXiv preprint arXiv:2501.05098 (2025).
>
> [9] Yang, Zhendong, et al. "Effective whole-body pose estimation with two-stages distillation." Proceedings of the IEEE/CVF International Conference on Computer Vision. 2023.
>
> [10] Bai, Shuai, et al. "Qwen2. 5-vl technical report." arXiv preprint arXiv:2502.13923 (2025).
>
> > Key elements of Orator seem to be incremental. Considering the amount of work involved in TalkCuts and the experimental results presented by the authors, this is not a key factor affecting my score.
>
> We thank the reviewer for the thoughtful feedback on the novelty and design of our approach. Below, we address the key concerns and highlight the contributions of our work.
>
> To begin with, the novelty and core contributions of our work lies in the following aspects: 1. **Novel Task and Dataset**: We define a new task: multi-shot speech video generation, a complex problem requiring the coordination of vocal delivery, gestures, and camera transitions. To support this, we created TalkCuts, the largest speech video dataset with over 500 hours of diverse content and rich annotations. This dataset lays a critical foundation for the research community; 2. **DirectorLLM as Orchestrator**: Our DirectorLLM provides fine-grained, interpretable guidance for camera transitions, gestures, and vocal delivery, seamlessly integrating with modular generation modules. This design balances flexibility, interpretability, and performance, addressing the limitations of naive end-to-end approaches.
>
> We acknowledge that the novelty in image animation components may appear incremental. However, please note that the task of multi-shot video generation is highly challenging and that we provide only a first baseline approach, as our main contribution is the novel dataset. As the first step in addressing the challenging task of multi-shot speech video generation, our pipeline lays a solid foundation for further exploration. Meanwhile, our modular system serves as an effective state-of-the-art solution for this task and provides a benchmark for future innovations.
>
> We sincerely appreciate your time for reviewing this paper and raising the valuable suggestions! If you have any further questions, feel free to let us know and we'll be happy to address them.
>
> Best,
>
> Authors

---

> > ### Comment · Reviewer_bXnd · 2025-08-06
> >
> > Thank you for the author's response. I acknowledge the novel task and dataset proposed in this work, and the author's rebuttal has resolved my doubts. Therefore, I maintain my positive score unchanged.

---

### Official Review · Reviewer_sRt1 · 2025-07-02

**Rating:** 5
**Confidence:** 4

**Summary:**

The submission includes a dataset for a new task of speech-driven video generation with multiple camera shorts spanning a range of scenarios. It contains richly annotated data for 500 hours of video. The annotations including camera transitions, video, audio, SMPL-X parameters, 2D keypoints, scene cuts, transcripts and video description. The authors include a baseline, Orator, which includes a DirectorLLM and Multimodal Video Generation module.
The efficacy of the dataset is seen in impressive results when used in a variety of tasks.

**Dataset Code Accessibility:**

Partly

**Dataset Code Comments:**

The dataset is accessible and the instructions for loading it are clear and well documented. However the code for accessing the model, more specifically, the line:
```
path = kagglehub.dataset_download("datasets/jiaben/talkcuts/versions/1")
```

runs into the following error
 "ValueError: Invalid dataset handle: datasets/jiaben/talkcuts/versions/1".

**Ethical Considerations:**

No, there are no or only very minor ethics concerns

**Final Justification:**

Thanks to the authors for engaging in the discussion and addressing the questions I had. The authors clearly addressed each of my concerns. Adding the information that I found missing/unclear would be valuable as the authors suggested. After the discussion, I will be sticking to the high rating of "5: Accept " as the paper has high impact, and introduces one of the first large datasets for Multishot Speech Video generation. I would hesitate to give a higher rating as I wouldn't fully call it groundbreaking, which is why I am happy to maintain my current rating of 5.

**Limitations Weaknesses:**

1. More clarity on how the camera shots are annotated would be better.
2. More details on how the voice instructions are annotated are needed.
2. As addressed in the paper, the dataset currently covers static environment.
3. Figure 4 needs more informative explanations. (For eg. Right hand seems to be missing for top right participant but the other hand is shown to indicate quality which is confusing).
4. Information about distribution of clip length per identity would be useful.

**Strengths Contributions:**

1. This paper introduces a large scale dataset that can be used to create Multi-shot Speech Video Generation Videos of humans.
2. The dataset has rich annotations that make it useful for a variety of Video Generation Tasks including LLM-guided camera shot transitions, audio driven human video generation and pose guided video generation.
3. The system is elegantly decomposed and provides controllability-  The Orator framework provides a way to use an LLM to plan changes in camera shots, voice delivery and the speaker's motion. This is then used to condition the Speech Generation and the Video Generation Models.
4. The efficacy of the dataset can be seen in the qualitative and quantitative analysis, where models trained on it seem to be beating existing baselines on shot coherence, motion quality, and identity preservation. The decomposition of these modules is possible because of the several layers of anotations provided in the dataset.
5. The dataset covers a large span of identities (10K) which is crucial for generalizability.
6. Comprehensive quantitative and qualitative evaluations are presented.

---

> ### Author Rebuttal · Authors · 2025-07-31
>
> Dear Reviewer,
>
> We thank the reviewer for the detailed reviews and insightful suggestions.
>
> > "More clarity on how the camera shots..."
>
> Thank you for your valuable suggestion. We provide a more detailed explanation of how the camera shot types are annotated below and will include a schematic diagram and detailed description in the supplementary material. We define six discrete camera shot types: Close-Up (CU), Medium Close-Up (MCU), Medium Shot (MS), Medium Full Shot (MFS), Full Shot (FS), and Wide Shot (WS), as shown in Part 1 of Figure 1 in the main paper. These annotations are determined automatically based on human keypoint visibility. Specifically, we use DWPose [1] to extract human keypoints. For each clip, we analyze the presence of keypoints corresponding to specific body parts and classify the shot type accordingly:
>
> - Close-Up (CU): At least one shoulder keypoint is visible
> - Medium Close-Up (MCU): At least one elbow keypoint is visible
> - Medium Shot (MS): At least one waist keypoint is visible
> - Medium Full Shot (MFS): At least one knee keypoint is visible
> - Full Shot (FS): All keypoints from head to feet are visible (full body within frame)
> - Wide Shot (WS): The full body is visible and the ratio of person height to image height is less than a particular threshold (0.7), indicating more background context.
>
> These rules ensure that each clip’s shot type reflects the speaker’s visible body region in a consistent and reproducible way. We will add a detailed illustration and annotation protocol in the supplementary material to improve clarity.
>
> [1] Yang, Zhendong, et al. "Effective whole-body pose estimation with two-stages distillation." Proceedings of the IEEE/CVF International Conference on Computer Vision. 2023.
>
> > "More details on how the voice instructions are annotated.."
>
> Thank you for the thoughtful comment. To clarify, our dataset does not contain annotated voice instructions. In our pipeline, the voice instruction which guides vocal attributes such as emotion, pace, and pitch is generated directly by prompting a LLM in a zero-shot manner, without any fine-tuning or retrieval-augmented generation. The LLM takes the input script as context and produces high-level vocal instructions, which are then used to condition the vocal generation. For vocal synthesis, we use CosyVoice [2], a state-of-the-art instruction-tuned TTS model. CosyVoice already supports rich zero-shot capabilities, including sentence-level control over emotion, speaking rate, and pitch via natural language instructions and token-level control for inserting elements such as laughter, breaths, and emphasis markers.
>
> Given the strong generalization ability of both the LLM and CosyVoice, we did not perform any additional fine-tuning, and thus no manually annotated voice instructions are required in our current pipeline.
>
> [2] Du, Zhihao, et al. "Cosyvoice: A scalable multilingual zero-shot text-to-speech synthesizer based on supervised semantic tokens." arXiv preprint arXiv:2407.05407 (2024).
>
> > "The dataset currently covers static environment."
>
> Thank you for pointing this out. As we understand, this refers to the fact that the backgrounds in our dataset are static, without dynamic scene changes or moving objects. We acknowledge this characteristic and would like to clarify that it stems from the nature of our target scenario: public speaking and presentation settings. Many of our videos are sourced from stage-based speech events, such as TED talks or talk shows, where the speaker remains the main focus and the background is naturally static. This setup reflects real-world constraints of the speech genre. Importantly, the static background environment is a key reason why relying on a single, fixed camera angle throughout a long speech can appear monotonous to viewers. This has led to the widespread use of dynamic camera shot transitions in real productions. Addressing this challenge and mimicking such transitions is one of the core motivations for the shot-planning task proposed in our dataset.
>
> > "Figure 4 needs more informative explanations..."
>
> Thank you for your insightful comment. We agree that Figure 4 requires a more informative explanation to avoid confusion. In the example you pointed out (top-right participant), we highlighted the left hand, where baseline methods produce noticeable artifacts, while our method yields significantly better results. Regarding the right hand, we acknowledge that our generated result is also imperfect in that area, however, the hand is not missing. It is facing directly toward the camera in that frame, and due to compression and resolution limits, it appears visually ambiguous.
>
> We also acknowledge that our current method does not explicitly address hand-level details. Multi-shot video generation remains highly challenging, and our work provides only a first unified baseline without component-level refinements. Even state-of-the-art methods like Hallo3 [3] exhibit similar hand artifacts. A promising direction for future improvement would be to leverage recent methods such as CyberHost [4], which introduce region-level attention modules and hand clarity scores to explicitly enhance hand fidelity.
>
> To clarify the purpose of the figure: Figure 4 is designed to highlight common artifacts in baseline outputs, including facial distortions, motion blur, mismatched hand gestures, and lip-sync inconsistencies, which are annotated with arrows and bounding boxes for clarity. Except for the specific example mentioned, our model consistently produces more visually realistic and temporally coherent results across all shot types, with improved fidelity in both facial and body regions. We will revise both the figure and caption to better explain these design choices and to avoid misinterpretations.
>
> [3] Cui, Jiahao, et al. "Hallo3: Highly dynamic and realistic portrait image animation with video diffusion transformer." Proceedings of the Computer Vision and Pattern Recognition Conference. 2025.
>
> [4] Lin, Gaojie, et al. "Cyberhost: A one-stage diffusion framework for audio-driven talking body generation." The Thirteenth International Conference on Learning Representations. 2025.
>
> > "Information about distribution of clip length per identity..."
>
> Thank you for the helpful suggestion. In Section 2.2 of the supplementary material, we have already included the total number of clips and shot size distribution per identity. However, we agree that presenting the distribution of clip lengths per identity is also important for a more complete understanding of the dataset. Each identity in our dataset is associated with approximately 3–5 minutes of raw speech video, which is segmented into multiple clips. The average clip length per identity is approximately 9.5 seconds, with a standard deviation of 4.2 seconds, and the majority of clips fall within the 5–15 second range. The distribution is relatively balanced across identities. We will update the supplementary material to include the detailed statistics and the analysis, along with corresponding histograms to visualize the per-identity clip length distribution.
>
> We sincerely appreciate your time for reviewing this paper and raising the valuable suggestions! If you have any further questions, feel free to let us know and we'll be happy to address them.
>
> Best,
>
> Authors

---

> ### Comment · Reviewer_sRt1 · 2025-08-08
>
> > These rules ensure that each clip’s shot type reflects the speaker’s visible body region in a consistent and reproducible way. We will add a detailed illustration and annotation protocol in the supplementary material to improve clarity.
>
> Thank you for clearly responding to this concern. The additional details make it much more clear and if included in the supplemental material, would be clearly address it well.
>
>
> > "More details on how the voice instructions are annotated.."
>
> Thank you for clarifying this concern too, I don't have additional comments for this since it has been addressed by the authors. I acknowledge the usefulness of this dataset and its novelty. I plan to maintain the original rating.
>
>
> > "The distribution is relatively balanced across identities. We will update the supplementary material to include the detailed statistics and the analysis, along with corresponding histograms to visualize the per-identity clip length distribution."
>
> Thanks for the details about the length of the clip per identity, that is an important detail and adding it to the supplemental material would be very helpful.
>
>
> The authors have addressed all my concerns.
>
> > "Many of our videos are sourced from stage-based speech events, such as TED talks or talk shows, where the speaker remains the main focus and the background is naturally static. "
>
> This explains the limitation of the dataset.
>
> > Except for the specific example mentioned, our model consistently produces more visually realistic and temporally coherent results across all shot types, with improved fidelity in both facial and body regions. We will revise both the figure and caption to better explain these design choices and to avoid misinterpretations.
>
>  Yes, this would be helpful. Thanks for acknowledging it.

---

### Note · Authors · 2025-08-13

Dear Reviewers and AC,

Thank you again for your time and valuable feedback. We are pleased that most concerns have been addressed, with three reviewers (NpwC, bXnd, sRt1) explicitly stated that their concerns were resolved in the rebuttal and that they will maintain their positive scores.

We appreciate that the reviewers recognize:
- The introduction of TalkCuts, a large-scale, diverse speech video dataset with rich multimodal annotations, broad identity coverage, and strong generalizability.
- The novel task of multi-shot speech video generation, and our “multi-role director” framework coordinating shot composition, body language, and vocal modulation.

For reviewer BTRq, we believe our rebuttal addressed their concerns, but since we have not received further feedback, we summarize the remaining points from reviewer BTRq and our responses below for completeness:

**Concern 1 — Use of separate reference images.**

Our baseline requires one reference image per shot. While effective as a first step, we agree an ideal system would synthesize novel shots from a single reference image. This remains unsolved for dynamic human videos, and TalkCuts provides a strong foundation to explore this in future work.

**Concern 2 — Comparison with camera-controlled methods & motion annotations.**

We clarify that camera motion and shot switching serve different roles; our work focuses on the latter, common in long-form speech. Direct comparisons are non-trivial, but our modular design can incorporate motion control. Around 10% of TalkCuts includes motion-annotated clips (via TRAM) to support future research.

**Concern 3 — Video results quality.**

Our baseline does not target fine-grained regions like hands, though similar artifacts appear in SOTA methods. While OmniHuman shows strong results, it uses a much larger private dataset with different conditioning. Using stronger backbones on TalkCuts already yields significant gains. Dataset quality is ensured via multi-stage filtering, human checks, and perceptual evaluation.

**Closing remark**

We believe these points address the remaining concerns and further demonstrate the soundness, validation, and research value of our framework and dataset. Thank you again for your time, attention, and consideration.

Best regards,

Authors

---

### Decision · Program_Chairs · 2025-09-18

**Decision:**

Accept (poster)

**Comment:**

This paper presents a large-scale dataset for studying multi-shot human speech video generation and an LLM-guided multi-modal generation framework as a simple baseline where the language model directs camera transitions, speaker gestures, and vocal modulation. Reviewers noted that compared with existing open-source datasets, the proposed dataset offers advantages in scale and provides clear definitions and classifications of different shots. They acknowledged the novelty of the task and dataset. The authors' rebuttal satisfactorily addressed the main questions and concerns raised by the reviewers.